# Landslide Displacement Prediction Based on a Two-Stage Combined Deep Learning Model under Small Sample Condition

Chunxiao Yu [1], Jiuyuan Huo [1,2,3,*], Chaojie Li [1] and Yaonan Zhang [2]

[1] School of Electronic and Information Engineering, Lanzhou Jiaotong University, Lanzhou 730070, China
[2] National Cryosphere Desert Data Center (NCDC), Lanzhou 730070, China
[3] Lanzhou Ruizhiyuan Information Technology Co., Ltd., Lanzhou 730070, China
[*] Correspondence: huojy@mail.lzjtu.cn; Tel.: +86-0931-4955743

**Abstract:** The widely distributed "Step-type" landslides in the Three Gorges Reservoir (TGR) area have caused serious casualties and heavy economic losses. The prediction research of landslide displacement will be beneficial to the establishment of local geological hazard early warning systems for the realization of scientific disaster prevention and mitigation. However, the number of observed data like landslide displacement, rainfall, and reservoir water level in this area is very small, which results in difficulties for the training of advanced deep learning model to obtain more accurate prediction results. To solve the above problems, a Two-stage Combined Deep Learning Dynamic Prediction Model (TC-DLDPM) for predicting the typical "Step-type" landslides in the TGR area under the condition of small samples is proposed. The establishment process of this method is as follows: (1) the Dynamic Time warping (DTW) method is used to enhance the small samples of cumulative displacement data obtained by the Global Positioning System (GPS); (2) A Difference Decomposition Method (DDM) based on sequence difference is proposed, which decomposes the cumulative displacement into trend displacement and periodic displacement, and then the cubic polynomial fitting method is used to predict the trend displacement; (3) the periodic displacement component is predicted by the proposed TC-DLDPM model combined with external environmental factors such as rainfall and reservoir water level. The TC-DLDPM model combines the advantages of Convolutional Neural Network (CNN), Attention mechanism, and Long Short-term Memory network (LSTM) to carry out two-stage learning and parameter transfer, which can effectively realize the construction of a deep learning model for high-precision under the condition of small samples. A variety of advanced prediction models are compared with the TC-DLDPM model, and it is verified that the proposed method can accurately predict landslide displacement, especially in the case of drastic changes in external factors. The TC-DLDPM model can capture the spatio-temporal characteristics and dynamic evolution characteristics of landslide displacement, reduce the complexity of the model, and the number of model training calculations. Therefore, it provides a better solution and exploration idea for the prediction of landslide displacement under the condition of small samples.

**Keywords:** "Step-type" landslide displacement prediction; small sample learning; data enhancement; two-stage combined deep learning model; parameters transfer

## 1. Introduction

### 1.1. Background

Landslides are severe geological disasters that directly cause casualties, house destruction, transportation network damage, etc., as well as a large number of secondary hazards such as debris flow caused by river blockage [1]. There are a number of landslide disaster-prone areas in China. Each year, landslide disasters have a massive impact on

human life and property, which are responsible for significant loss of life and injury to people and their livestock, as well as damage to infrastructure, agricultural lands, and housing. Early identification, prediction, and early warning research based on remote sensing (RS) and the global position system (GPS) of landslides can effectively prevent geological disasters and avoid many casualties and property losses [2].

The Three Gorges Reservoir (TGR) is one of China's most important water conservancy facilities. It not only undertakes the function of power generation but also plays an important role in flood control, shipping, and water supply [3]. "Step-type" landslides are widely distributed in this region, which all show a curve form of stepwise displacement [4]. However, the samples of GPS stations used to monitor landslide displacement in the TGR area are small in monthly units, so the internal mechanism of landslide displacement analysis cannot be fully excavated. This paper will propose a reliable and accurate displacement prediction method under the condition of small samples of observation data.

*1.2. Related Work*

The traditional extraction and prediction method of landslide disaster mainly rely on artificial visual interpretation. This method relies heavily on the accuracy of manual measurement methods and the experience of landslide prediction experts, which is time-consuming and labor-intensive and has significant limitations [5]. Researchers have proposed some predictive analysis methods based on physical and data models in recent years [6,7]. Among them, it is very challenging to establish a fully representative physical model in the research because the displacement process and corresponding mechanism process of landslide disasters are very complex.

The landslide displacement prediction method based on the data model mainly analyzes the curve of time vs. landslide displacement and the monitoring information of various external factors such as rainfall, reservoir water level, etc. Firstly, the nonlinear relationship between external influencing factors and landslide displacement is captured to invert the inherent nonlinear dynamic evolution process of landslide. Then, the evolution trend of landslides can be predicted by extrapolation [8]. The models established in this way do not need to pay too much attention to the mechanical process of landslides, but they have the advantage of higher prediction accuracy. Therefore, prediction methods based on data models have gradually become the main focus of current research on landslides and other geological disasters. Traditional landslide displacement prediction methods based on data models mainly include the Verhulst model [9], grey model [10], exponential smoothing model [11], etc. Nandi et al. selected a variety of landslide instability-inducing factors, including landslide angle, nearby water flow, soil type, and soil erodibility, etc., and then analyzed the multivariate statistical relationship between landslide displacement and instability factors by logistic regression method and established the correlation between factors and landslide displacement changes [12]. However, in these studies, due to the non-linear relationship between various instability-inducing factors and landslide displacement changes, the data-based prediction model cannot accurately describe the causal relationship between variables. Therefore, these models can only be effective for landslide prediction with similar deformation characteristics under specific conditions, but they have significant limitations in predicting widespread and scattered landslides.

In recent years, with the further development and application of Machine Learning algorithms, landslide displacement prediction modeling based on machine learning methods has been widely studied and applied. Zhu et al. used a Back Propagation (BP) neural network optimized by Genetic Algorithm (GA) and Particle Swarm Optimization (PSO) algorithm to assess geological hazard risk [13]. Fan et al. predicted landslide displacement of the Baishuihe River by variational modal decomposition and AMPSO-SVM coupling model [14,15]. Based on the response analysis of impact factors, Bui et al. used the translation index method to decompose the cumulative landslide displacement into different components and then built an Artificial Neural Network (ANN) model to predict the components [16]. The models based on Support Vector Machine (SVM) and Neural

Network (NN) used in these studies are static models. The static models ignore the important feature that landslide evolution is a complex dynamic evolution process and treat it as a simple static regression problem. Therefore, when the data of the key influencing factor changes dramatically, the model cannot capture the evolution characteristics of the influencing factor over time, which results in the model's prediction accuracy being heavily limited [17].

Compared with the limitations of static models in prediction, the dynamic deep learning model can better reflect the dynamic evolution process of landslides and improve the prediction accuracy of landslide displacement [18]. Prediction models based on Recurrent Neural Networks (RNNs) are different from traditional Artificial Neural Network (ANN) methods in that they can use internal memory units to process arbitrary input sequences to learn and understand the features of time series. The internal nodes of the RNN model are connected recursively so that the current node can remember the historical information and realize the state feedback of the network [19]. Long Short-Term Memory (LSTM) network is a variant of the RNN network, and its unique RNN network structure can well control the proportion of long-time memory and short-time memory through gating and solve the problems of gradient disappearance and the explosion of RNN, so it is favored by many researchers [20]. In landslide displacement prediction, LSTM also achieves higher prediction accuracy than SVM, Elman, and other models [21].

Although Deep Learning methods have been applied in many fields and achieved many excellent research results, these methods require a large number of or even massive sample data in the process of training and modeling. A small amount of training data cannot build a robust and accurate deep learning model, so it cannot reflect the models' advantages in prediction application. To enable the model to acquire the ability to learn and generalization as much as possible from a small number of samples, researchers proposed a small sample learning method, which can solve the problem of insufficient model training due to a small number of samples by using prior knowledge in data, model and algorithm [22]. In terms of data, prior knowledge can be used to enhance supervisory information. For instance, Schwartz et al. set up a series of autoencoders for learning in similar categories and then added the learned changes to the original sample as a new sample [23]. Kwitt et al. used the attribute strength regressor learned from large-scale data to transform a single sample into several samples while keeping the original labels unchanged [24]. Wu et al. used a progressive strategy to select valuable unlabeled samples and assign a false label to them for CNN network training [25]. Le et al. proposed window slices to randomly extract sub-samples of continuous slices from the original time series to generate new time-series data [26]. Hassan et al. enhanced the data of sequential data through the center-weighted average method based on the Dynamic Time Warping (DTW) algorithm and finally achieved better results in the classification of sequential data [27]. In terms of the model, prior knowledge can be used to reduce the size of the hypothesis space. For example, Zhang et al. made the networks of the two tasks share the parameters of the first several layers to extract general information, and the last layers of the network learned and updated parameters respectively to adapt to different outputs [28]. Motiian et al. trained a variant autoencoder from the source task and copied it to the target task, where the two autoencoders shared some layers for capturing general information, and the target task only updated its proprietary layer parameters [29].

### 1.3. Article Arrangement

The above-mentioned small sample learning methods have made significant progress in the field of computer vision, such as image classification, object detection, case segmentation, etc. However, there are still few applications in research on time series prediction, landslide susceptibility analysis, and landslide displacement prediction based on small samples of GPS monitoring data. Therefore, the main contributions of this paper are as follows:

1.  A decomposition method based on first-order and second-order difference DDM (Difference decomposition method) is proposed for landslide displacement decomposition.
2.  A data enhancement method based on DTW (Dynamic Time Warping) method is proposed to enhance the few-shots GPS samples.
3.  A model enhancement method based on a two-stage CNN-attention-LSTM combined deep learning model named TC-DLDPM was built to extract the high non-linearity and complexity of spatial and temporal correlations in landslide displacement.

The rest of this paper is organized as follows. Section 2 describes the suggested materials and methodology. The study area and dataset are described in Section 3. The model implementation is introduced in Section 4. The comparative experiments and correlation analysis are discussed in Section 5, and finally, the conclusions and potential future works can be obtained in Section 6.

## 2. Materials and Methods

In this paper, a Two-stage Combined Deep Learning Dynamic Prediction Model (TC-DLDPM) is proposed for the prediction of cumulative landslide displacement under small samples condition. The overall prediction flow chart is shown in Figure 1, and the specific process is as follows:

(1)  According to the Difference Decomposition Method (DDM), the cumulative landslide displacement is decomposed into trend displacement and periodic displacement components.
(2)  For the trend displacement component, the cubic curve fitting method is adopted for fitting and modeling in this paper to realize the trend prediction model of the displacement.
(3)  For the periodic displacement component, firstly, through analysis and evaluation of the related external factors which induced landslides, the displacement data of the monitoring station similar to the target dataset is fused together. Then, the data enhancement method DTW algorithm in the small sample learning is used to enhance the base dataset and the target datasets, respectively, to build an extensive base dataset.
(4)  Train the TC-DLDPM prediction model on the base and target datasets, and finally obtain the prediction result of periodic displacement components on the target testing dataset.
(5)  The predicted values of the trend displacement component and periodic displacement component are superimposed, and finally, the cumulative displacement prediction result can be obtained.
(6)  Compare and analyze the cumulative displacement prediction results with the observed data to evaluate the efficiency and performance of the model.

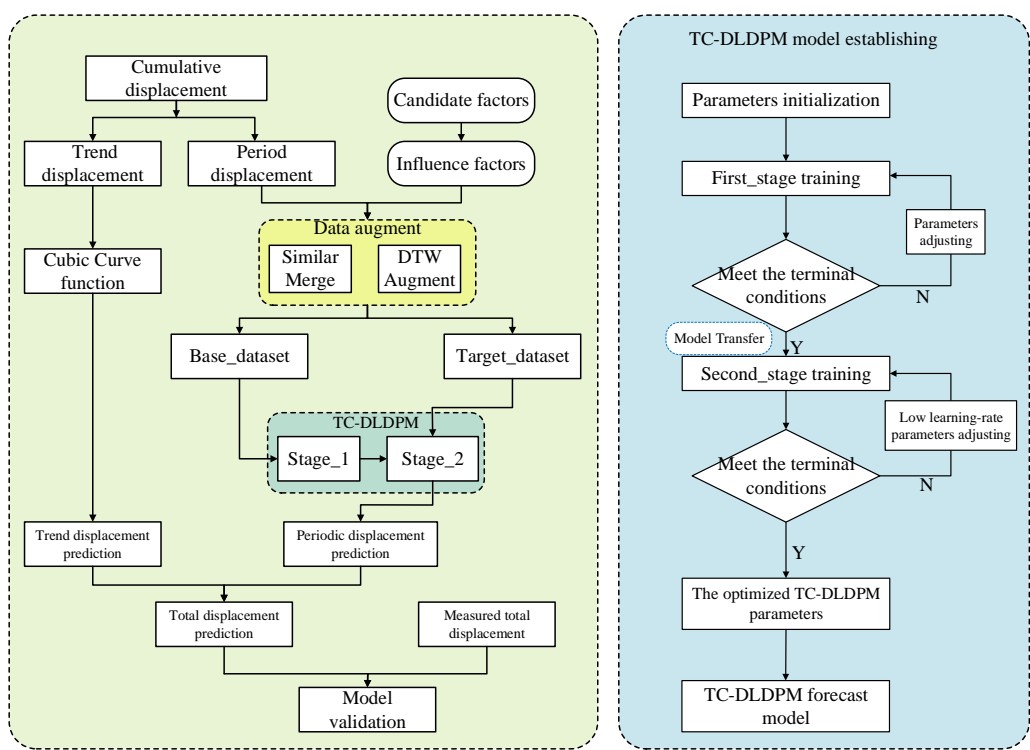

**Figure 1.** Flow chart of cumulative landslide displacement prediction.

### 2.1. Difference Decomposition Method (DDM) for Landslide Displacement Decomposition

The formation and occurrence of landslide is a very complex evolutionary process, which is affected by the internal factors of the slope body, such as stratum lithology, landslide fluid pressure in pores, gravity stress, etc., and external trigger factors such as rainfall, earthquake, human activities, vegetation coverage, water level fluctuation, etc. The landslide displacement under the effect of internal factors of the slope body presents an approximately monotonic increasing curve, which is called trend term displacement. The landslide displacement affected by external trigger factors is periodic, which is called periodic term displacement. Therefore, the cumulative landslide displacement can be regarded as the superposition of the trend term displacement component and the periodic term displacement component. Formula (1) is the cumulative landslide displacement decomposition mode [30].

$$S(t) = \varnothing(t) + \eta(t) \tag{1}$$

In which, $S(t)$ represents the cumulative landslide displacement time-series, $\varnothing(t)$, denotes the displacement component of the trend term that is influenced by the slope itself, and $\eta(t)$ represents the displacement component of the periodic term affected by external factors such as rainfall and reservoir water level change.

This study uses the difference decomposition method to decompose the cumulative landslide displacement into trend term displacement and periodic term displacement components. The difference decomposition method selects the "key" points of the trend term displacement according to the first-order and second-order difference values of the cumulative landslide displacement data. These "key" points are the step mutation points of the "Step-type" landslide, and their physical meaning is the critical point of the change of the slope body's own geological conditions. After finding these key points, the cubic spline interpolation method is used to complement the entire trend displacement component, and finally, the trend displacement component sequence is obtained. The periodic displacement component value equals the cumulative component displacement value minus the trend displacement component value.

The specific implementation process of the difference decomposition method is as follows: firstly, the first-order difference sequence $V$ and second-order difference sequence $A$ of cumulative landslide displacement data are calculated. The calculation formulations of $V$ and $A$ are shown in Formulas (2) and (3), respectively.

$$V = \{v_1, v_2, \ldots, v_n | v(t) = s(t) - s(t-1)\} \qquad (2)$$

$$A = \{a_1, a_2, \ldots, a_n | a(t) = v(t) - v(t-1)\} \qquad (3)$$

In which, $S(t)$ represents the cumulative landslide displacement value at the current moment, $V(t)$ represents the first-order difference value of the displacement at the current moment, and $A(t)$ represents the second-order difference value of the displacement at the current moment.

According to the $V(t)$ and $A(t)$ of the landslide, the "key" points of the trend displacement component are selected, which can make the trend term displacement component behave as an approximately monotonic increasing function with a large time scale, and the periodic term displacement component presents a periodic curve. Therefore, the displacement component of the trend item can fully reflect the long-term change trend of the slope under its own geological conditions.

The flow chart of the DDM method of the displacement decomposition is shown in Figure 2. When $V_i$ at time $t$ is both smaller than that at time $t-1$ and $t+1$, or $V_i$ at time $t$ is less than 0 and the $A_i$ is greater than zero, indicating that time $t$ is the critical point for the step change of landslide displacement. The critical points of these step changes are extracted as the "key" points of the trend displacement component, and the value of all time is completed by the cubic spline interpolation method; then, the trend displacement component with an approximate monotone increasing curve can be obtained. These "key" points show that when the slope's geological conditions change, it will lead to the step mutation of landslide displacement. The displacement component value of the periodic term is the cumulative landslide displacement value minus the trend term component value. The displacement component value of the periodic term shows reasonable periodicity by eliminating the step mutation caused by the slope's own geological conditions.

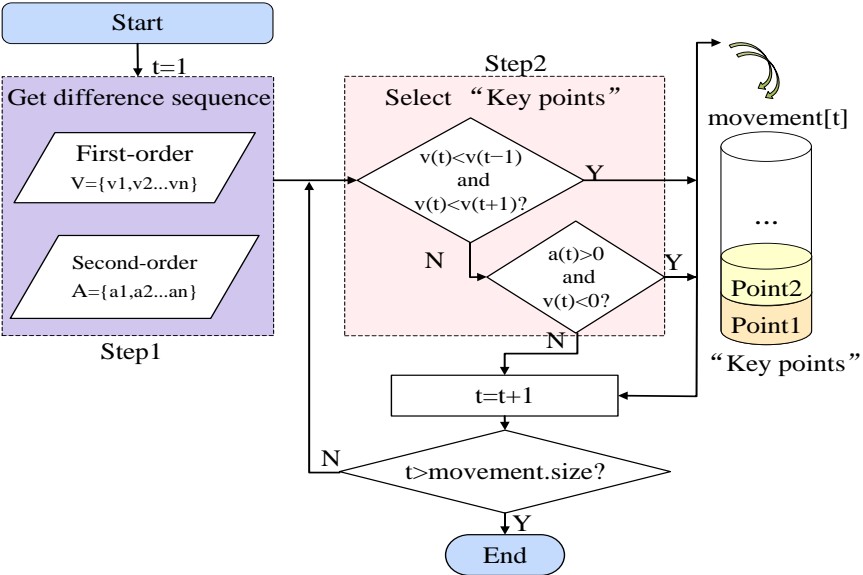

**Figure 2.** Flow chart of DDM method.

Therefore, the differential decomposition method can decompose the cumulative landslide displacement well, and the monotone increasing trend term and the periodic term displacement component can be obtained. Compared with Empirical Mode Decomposition (EMD) [31] and Moving Average (MA) [32], the displacement component of trend term

decomposed by this method contains step mutation points under its own geological conditions. It is more consistent with the physical meaning of the displacement component of the trend term and can get the displacement component of the periodic term with stronger periodicity, which is conducive to the subsequent mining of the inner evolution rule of the periodic displacement component, thus improving the prediction accuracy.

*2.2. Enhancement Method Based on DTW of GPS Data*

To learn the general Spatial-temporal characteristics of the "Step-type" landslide displacement, the deep learning model first needs to be trained on a large dataset that is similar to the characteristics of the small sample dataset, and after training the parameters of the migration to the small sample data, thus narrow the hypothesis space and achieve a more accurate capture of the input-output relationship of the target dataset. Therefore, before building the dynamic prediction model, it is necessary to construct a base dataset with a large sample size so that the dynamic prediction model can be fully trained on this dataset. Then, the general characteristic parameters of the "Step-type" landslide learned from the base dataset are transferred to the training of the target dataset to solve the problem of insufficient model training due to the small sample size of the target data set to a certain extent. To ensure that the universal feature parameters trained on the base dataset can be successfully transferred to the target dataset and the prediction accuracy can be improved, there should be a high degree of similarity between the base dataset and the target dataset. In this paper, data from multiple monitoring stations that are similar in geographic environment to the target dataset and their monitor landslides of the same type as "Step-type" landslides are selected for fusion to construct a large base dataset.

Data enhancement on a small sample dataset is a relatively simple and effective solution. Data enhancement is also called data augmentation, which refers to adding new data to the original small sample dataset with the help of auxiliary data or transformation rules to generate a more extensive dataset and prevent the problem of over-fitting in model training. In this paper, the center-weighted average method based on Dynamic Time Warping (DTW) [33] is used to enhance the stability and robustness of the base dataset and target dataset, respectively. DTW algorithm is essentially a dynamic programming algorithm for calculating the similarity of two-time series. It is mainly used for time series data enhancement, including isolated word speech recognition, gesture recognition, data mining, and information retrieval. The algorithm first needs to define template sequence $C$ and query sequence $Q$ with lengths $m$ and $n$, respectively, where $C = c_1, c_2, \ldots, c_j, \ldots, c_m$, $Q = q_1, q_2, \ldots, q_i, \ldots, q_n$. Then a matrix of size $n \times m$ is constructed, and the matrix element $(i, j)$ represents the cumulative distance $\gamma(i, j)$ from the origin $(Q_1, C_1)$ to $(Q_i, C_i)$. The DTW algorithm can find the shortest path $\gamma(n, m)$ from the origin $(Q_1, C_1)$ to the endpoint $(Q_n, C_m)$ of two sequences. The shortest path $\gamma(n, m)$ is the similarity $S$ between sequence $Q$ and sequence $C$. The calculation process of using the DTW algorithm to obtain the cumulative distance $\gamma(i, j)$ of the point $(Q_i, C_i)$ is shown in Formulas (4)–(7) [34].

$$\gamma(0, 0) = d(0, 0) \tag{4}$$

$$\gamma(i, 0) = d(i, 0) + \gamma(i - 1, 0); i \geq 1, j = 0 \tag{5}$$

$$\gamma(0, j) = d(0, j) + \gamma(0, j - 1); i = 0, \ j \geq 1 \tag{6}$$

$$\gamma(i, j) = d(i, j) + \min\{\gamma(i - 1, j), \gamma(i, j - 1), \gamma(i - 1, j - 1)\}; i \geq 1, j \geq 1 \tag{7}$$

where Formula (4) represents the cumulative distance of the origin, Formula (5) represents the cumulative distance of the elements in the first column of the matrix, Formula (6) represents the cumulative distance of the elements in the first row of the matrix, and Formula (7) represents the calculation formula of the cumulative distance of other elements in the matrix. The distance between the two points of the sequence $Q$ and $C$ can be calculated by $d(i, j) = (Q_i - C_j)^2$, that is, the Euclidean distance between the two points.

The new sample generation process based on the DTW algorithm is shown in Figure 3. Firstly, a sample is randomly selected from the original data set as the template sequence $C$. Then, 30 samples were selected as query sequences, and the DTW algorithm was used to screen out 5 query sequences $Q_1 - Q_5$ with the highest similarity to template sequences. According to the rules shown in Figure 2, different weights were allocated for samples $C$, $Q_1$, $Q_2$, $Q_3$, $Q_4$, and $Q_5$. Finally, a new sample was generated through weighted summation and added to the original data set.

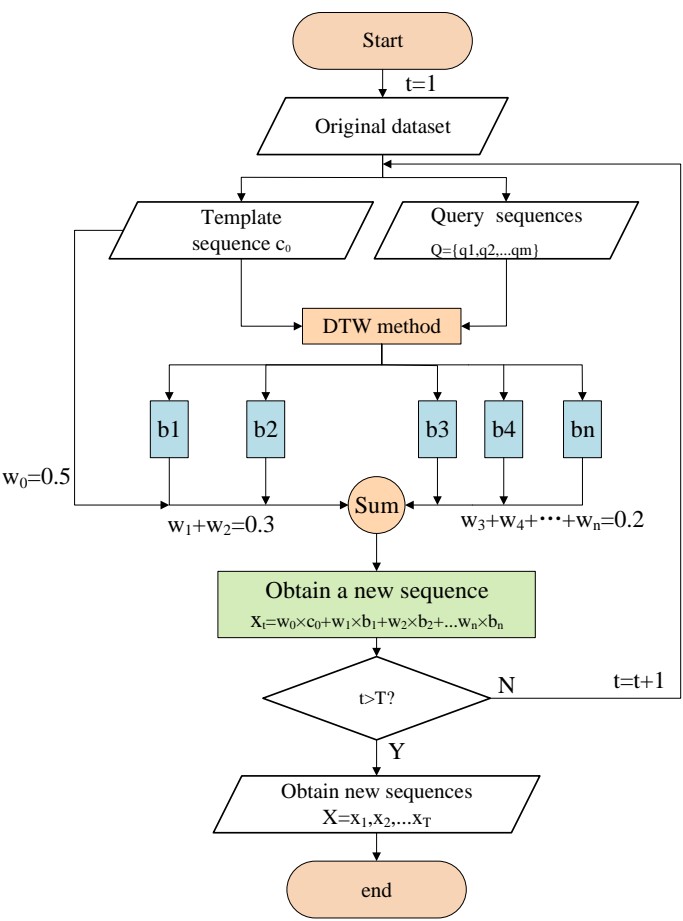

**Figure 3.** Data enhancement process based on the DTW algorithm.

By weighting and averaging time series samples with high similarity to generate new time series samples, the data enhancement method based on the DTW algorithm can make the enhanced and expanded data set have better stability and robustness. At the same time, the prediction model based on deep learning can be more thoroughly trained on the enhanced data set and better dig out the relationship between the input and output to obtain higher prediction accuracy on the small data set that originally shows an over-fitting problem.

### 2.3. Two-Stage Combined Deep Learning Prediction Model TC-DLDPM

The specific structure of the basic model of TC-DLDPM is shown in Figure 4. The model supports long input sequences and splits the input sequences into timestep-based sub-sequences. One-dimensional convolution and pooling operations on each sub-sequence can effectively reduce the scale of features, thereby extracting valid information and improving computational efficiency; then assigning different attention to the timing association and influencing factor association, respectively, through the Timestep-Features Attention block, and stitching the computed subsequence as the input of the LSTM network. Finally, through the LSTM network training, accurate predictions are obtained. The TC-DLDPM

model can capture the spatio-temporal characteristics and dynamic evolution characteristics of landslide displacement, reduce the complexity of the model, and the number of model training calculations, and at the same time, this paper introduces an attention mechanism to learn and adjust the heterogeneity parameters of spatio-temporal features, so that the model pays different degrees of attention to each input feature at different times. The working principles and functions of each network layer are described as follows:

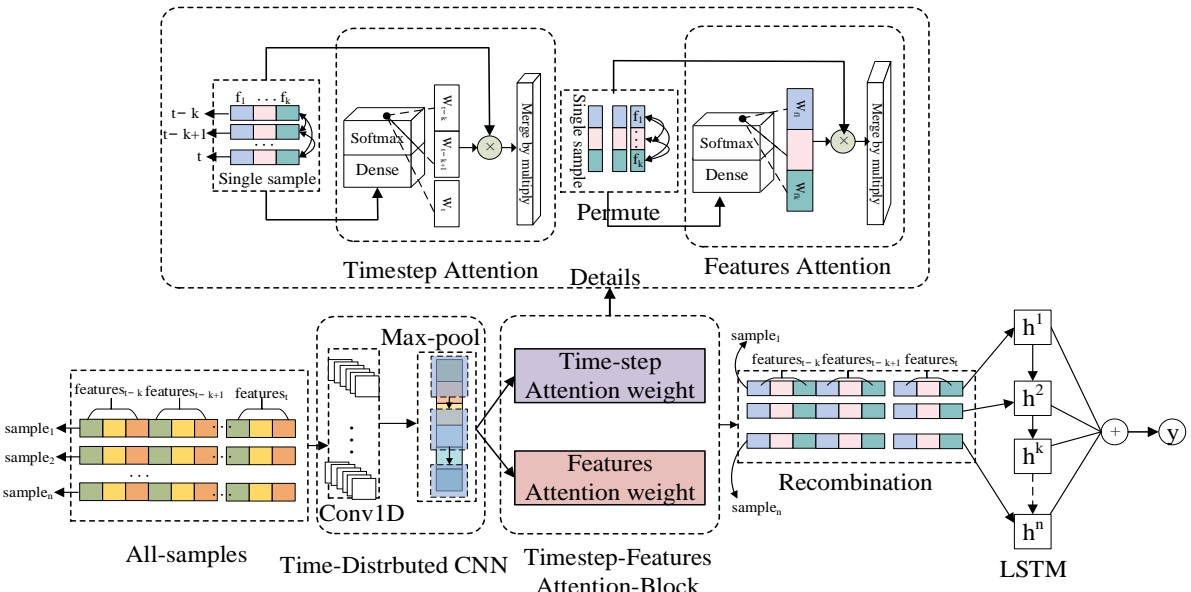

**Figure 4.** The basic model of TC-DLDPM.

1.    CNN neural network

The CNN neural network can extract significant Spatio-temporal features from the input data while reducing the number of network parameters and reducing the number of model calculations [35]. CNN neural network is mainly divided into convolution layer and pooling layer, which realize convolution operation and pooling operation on input data, respectively. The calculation formula of the one-dimensional convolution layer is shown in Equation (8) [36]:

$$y_c = \sigma(W_c \otimes X + b_c) \tag{8}$$

In which, $y_c$ represents the output sequence of the convolution layer, $W_c$ represents the weight value of one-dimensional convolution kernel, $b_c$ is the bias vector, $\otimes$ is convolution operation. $\sigma(.)$ the activation function of the convolution layer and the $ReLU(.)$ function is taken in this paper.

To further reduce the complexity of the model and the amount of calculation, the output sequence $y_c$ which after the convolution operation needs to be taken as the input of the maximum pooling layer. The calculation formula of the maximum pooling layer is shown in Formula (9).

$$y_p = \max_{r \in R} y_{i \times T + r} \tag{9}$$

In which $T$ represents the pooling step, $R$ represents the size of the pooling window, and $r$ represents the offset within the pooling window.

2.    Attention layer

The main function of the Attention Mechanism is to allocate computing resources to more important training tasks when computing power is limited. The attention mechanism can selectively allocate more attention to important information and ignore most of the less important information by imitating people's way of perception [37]. In processing a large amount of input information by the neural network model, the attention mechanism

can be used to select only some key input information for processing, and the most concerned information at the current time step can be obtained by assigning weights weighted summation. The calculation steps of the attention value are: (1) Calculate the attention distribution on all input information; (2) Calculate the weighted average of the input information according to the attention distribution. The definition of attention weight value is shown in Formula (10).

$$W = \left( W^1, W^2, \ldots, W^L \right) \tag{10}$$

Based on the attention weight value, the importance of the input data sequence can be sampled. The sampling formula is shown in Formula (11). By sampling the significance of the input data sequence, the network can assign different attention degrees to each input data.

$$\widetilde{X_t} = \left( x_t^1 W^1, x_t^2 W^2, \ldots, x_t^L W^L \right) \tag{11}$$

In which, $X_t = \left( x_t^1, x_t^2, \ldots, x_t^L \right)$ represents the input time series data at time $t$. $\widetilde{X} = \left( \widetilde{X_1}, \widetilde{X_2}, \ldots, \widetilde{X_T} \right)$ represents the sequence data weighted by the attention weight value.

In this paper, the timing correlation and environmental characteristic association of the input data are calculated separately. As shown in the network details of the Timestep-Features Attention block in Figure 3, the model assigns an initial weight vector WT ($W_T = \{ w_{t1}, w_{t2,\ldots,} w_{tn} \}$) to each sub-sequence based on the time step split firstly, and obtains the correlation between these sub-sequences through training; then transposes the sub-sequence and assigns the initial weight vector WF ($W_F = \{ w_{f1}, w_{f2,\ldots,} w_{tm} \}$) to different environmental features in the sequence, different correlations between features are obtained through attention calculation. The above method focuses on the important information from the two dimensions of timing and environmental characteristics, which makes the extraction of effective information more detailed and accurate.

3.　Long and Short-Term Memory (LSTM) neural network

LSTM is a variant RNN network, and its network structure is shown in Figure 5. In this model, mechanisms such as memory cell, input gate, forget gate, and out gate are added based on the RNN network. Compared with the RNN network, cell state c is newly added to the LSTM model to store the long-term state. LSTM uses an input gate and a forget gate to control the contents of cell state C. The input gate determines the amount of information that the current network input $x_t$ saves to the unit state $c_t$, the forget gate determines the amount of information saved in the cell state $c_{t-1}$ at the previous step in the cell state $c_t$. The output gate controls the amount of information output from the cell state $c_t$ to the current output $h_t$ of the LSTM. By adding the above-mentioned gate control mechanism, LSTM can control the proportion of long-term memory and short-term memory and solve the problems of gradient disappearance and gradient explosion of RNN.

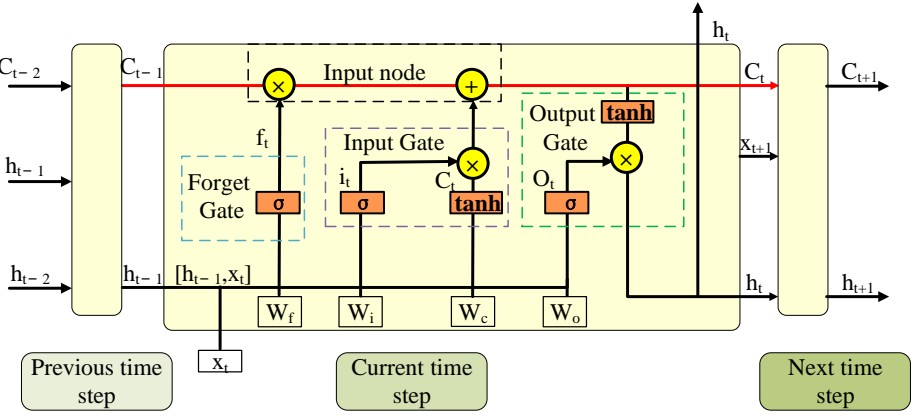

**Figure 5.** LSTM network structure.

The corresponding functions of input gate, forgetting gate, and output gate mechanisms are shown in Formulas (12)–(14), respectively [38].

$$I_t = \sigma(W_{i1}x_t + W_{i2}h_{t-1} + b_i) \tag{12}$$

$$f_t = \sigma\left(W_{f1}x_t + W_{f2}h_{t-1} + b_f\right) \tag{13}$$

$$O_t = \sigma(W_{O1}x_t + W_{O2}h_{t-1} + b_O) \tag{14}$$

where $I_t$, $f_t$, and $O_t$, respectively represent the values of input gate, forget gate, and output gate of LSTM neural network at time $t$. $x_t$ represents the input at time $t$, $h_{t-1}$ represents the output of the LSTM network at time $t-1$. $W_1$ is the connection weight between the input node and the hidden layer node, $W_2$ represents the connection weight between nodes of the hidden layer and output nodes. $b_i$, $b_f$, and $b_o$ are the offset items corresponding to the input gate, forget gate, and output gate, respectively.

As shown in Figure 6, the TC-DLDPM realizes the learning and training of the base dataset in stage 1 (Basic training) to obtain the base displacement prediction model. Then, the CNN and LSTM layers' parameters in stage 1 are retained in stage 2 (Small sample fine-tuning) to realize the learning and training of the target dataset. The displacement prediction model of the real target dataset is established. The model training process is divided into two stages, basic training and small sample fine-tuning, to achieve accurate landslide displacement prediction under the condition of a small sample. The specific process steps are as follows:

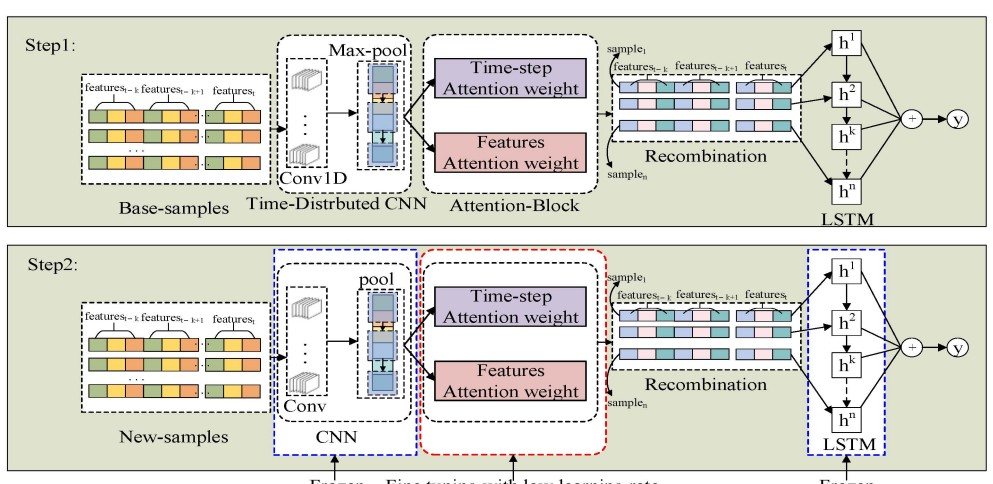

**Figure 6.** The training process of the TC-DLDPM model.

The base dataset and the target dataset are, respectively, enhanced by the center-weighted average method based on the Dynamic Time Warping (DTW) algorithm to construct a base dataset with a large sample size. After the base data set is preprocessed, it is input into the combined deep learning model.

Based on the CNN network, LSTM network, and attention mechanism, the stage 1 learning and training on the base dataset are studied and trained, and the base displacement prediction model is constructed. In addition, the base model has obtained the network parameters of each layer and the prediction capability.

Migrate the parameters of the base model to the target dataset under the condition of small samples, freeze the LSTM and CNN layers' parameters in the network, then fine-tune the Time-step&features Attention-block layers' parameters at the 1/20 learning rate of the original base model in stage 2, finally obtain the TC-DLDPM with better stability and accuracy.

*2.4. Evaluation Indicators of Model Performance*

The indicators of Root Mean Square Error (RMSE), Mean Absolute Error (MAE), and correlation coefficient (R) are selected to evaluate the predictive performance of the proposed TC-DLDPM. At the same time, absolute error, relative error, and model prediction time are also taken as the analysis indicators of prediction results.

The calculation formulas of RMSE, MAE, and R are shown in Formulas (15)–(17), respectively.

$$\mathrm{RMSE} = \sqrt{\frac{\sum_{i=1}^{n}(x_i - \hat{x}_i)^2}{N}} \tag{15}$$

$$\mathrm{MAE} = \frac{\sum_{i=1}^{n}(x_i - \hat{x}_i)^2}{N} \tag{16}$$

$$\mathrm{R} = \frac{\sum_{i=1}^{n}(x_i - \overline{x})(\hat{x}_i - \overline{\hat{x}})}{\sqrt{\sum_{i=1}^{n}(x_i - \overline{x})^2}\sqrt{\sum_{i=1}^{n}(x_i - \hat{x}_i)^2}} \tag{17}$$

In which, $x_i$ and $\hat{x}_i$ represent the real value and predicted value, and $\overline{x}$ and $\overline{\hat{x}}$ represents the average value of the real and predicted landslide displacement values, respectively. $N$ is the number of samples.

## 3. Study Area and the Landslide Displacement Data Set

The TGR area is a narrow, steeply sloped area that runs along the middle reaches of the Yangtze River and flows through large swaths of limestone mountains. With the impaction of the impoundment and the surrounding human activities, it has been one of the most landslide-prone areas in China. This paper mainly focuses on Baishuihe and Bazimen, two serious landslide sites in this area, and collects displacement, rainfall, and reservoir water level data from the above two monitoring stations for the composition of subsequent experimental data sets.

### 3.1. Study Area

The study area of this paper is the Three Gorges Reservoir area of China. The Three Gorges Reservoir area stretches from Wanzhou District of Chongqing Municipality in the west to Yichang City of Hubei Province in the east, about 380 km in length. The reservoir area is characterized by numerous mountains, steep terrain, ravines, and rivers. The reservoir flows through large limestone mountains, providing physical conditions for the formation of landslides, thus leading to frequent geological disasters in this area [39]. According to the geological report of the Three Gorges Region released by the China Geological Survey in 2019, there are about 7926 geological disasters occurred in the area, including landslides, collapse, debris flow, and ground collapse. Among them, 6814 landslides (2 giant ones, 88 super large ones, and 1094 large ones) are mainly distributed on the slopes of the Yangtze River and its tributaries with an elevation of 200 m to 100 m and are primarily developed on the slope areas with weak geological conditions such as clastic rocks, intense human engineering activities, river intersection and intense geological tectonic activities [40]. Therefore, landslides have become the most serious geological disaster in the Three Gorges Reservoir area. At the same time, the periodic rise and fall of the water level of the Three Gorges Reservoir and the periodic changes in rainfall have also triggered a series of new landslide disasters, which brought substantial economic losses and casualties. In 2003, the Qianjiangping landslide in the Three Gorges Reservoir area caused 24 deaths, destroyed 4 factories and 129 houses, overturned ships, blocked the river, and caused heavy losses [41]. Therefore, it is of great significance to carry out landslide prediction and control in the Three Gorges area.

There are a wide range of "Step-type" landslides in the Three Gorges Reservoir area, which exhibits a curve form of displacement step by step. As shown in Figure 7, this paper conducts displacement prediction research on such "Step-type" landslides, focusing on

the analysis of landslide displacement data in the Baishuihe area and the Bazimen area in the Three Gorges reservoir area. Among them, the Bazimen landslide is located at the mouth of the right bank of the Xiangxi River, a tributary of the Yangtze River in Guizhou Town, Zigui County, Hubei Province, 31 km away from the Three Gorges Dam. The geographical coordinates of the landslide are 110°45′30″ Longitude and 30°58′16″ Latitude. The landslide belongs to the accumulation layer landslide, and the distribution elevation of its body is from 139 m to 280 m. The landslide is 350 m in length, 350–500 m in width, 30 m in average thickness, and about 4 million m³ in volume. Four Global Positioning System (GPS) deformation monitoring stations (ZG111, ZG110, ZG112, and ZG109) are deployed in the Bazimen area. Another landslide, the Baishuihe landslide, is located in Zigui County of the Three Gorges Reservoir area, 56 km away from the Three Gorges Dam. The geographical coordinates are 110°32′09″ Longitude and 31°01′34″ Latitude. The landslide is located in the broad valley section of the Yangtze River, and the monocline bedding slope is high in the south and low in the north, spreading towards the Yangtze River in a ladder shape. Its length from north to south is 600 m, and its width from east to west is 700 m. The average thickness of the slide body is about 30 m, and the volume is 1260 × 104 m³. It belongs to the type of large accumulation landslide. According to the topography, geological conditions, deformation characteristics, and observation accessibility of the landslide, 11 GPS deformation monitoring stations (XD01, XD02, XD03, XD04, ZG91, ZG92, ZG93, ZG94, ZG118, ZG119, ZG120) have been deployed by relevant experts in Baishuihe Area, which is shown in Figure 8 [42].

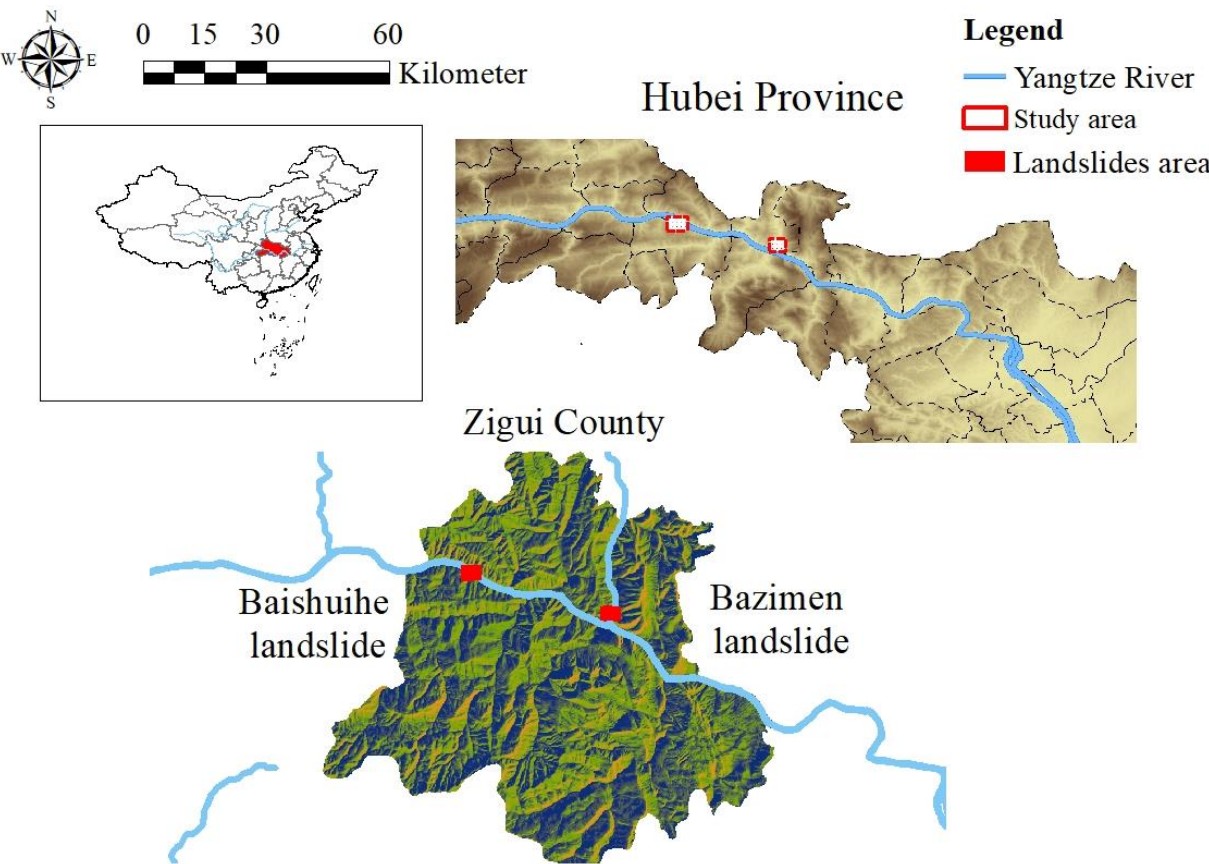

**Figure 7.** Location map of the study area.

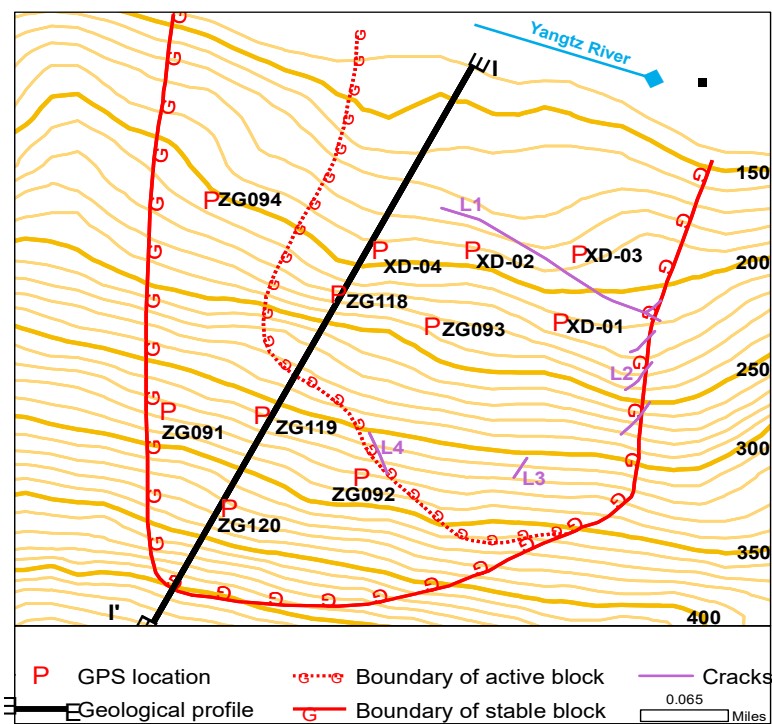

**Figure 8.** The Deployment location of GPS monitoring points.

The deformation of geological disasters in the Three Gorges Reservoir area is mainly divided into oscillation, linear, and step types. The step type is the main deformation form of geological disasters with obvious deformation and relatively obvious deformation, which is the key type of geological disaster deformation in the reservoir area that needs to be monitored. As shown in Figure 9, the step type indicates that the cumulative displacement-time curve has an obvious step in a certain period of time, showing the characteristics of step rise. The Baijiaobao landslide and Baishuihe landslide in Zigui County belong to this kind of step landslide, with annual deformation generally ranging from tens of millimeters to hundreds of millimeters, and in the state of deformation or obvious deformation, or even greater or unstable failure damage [43].

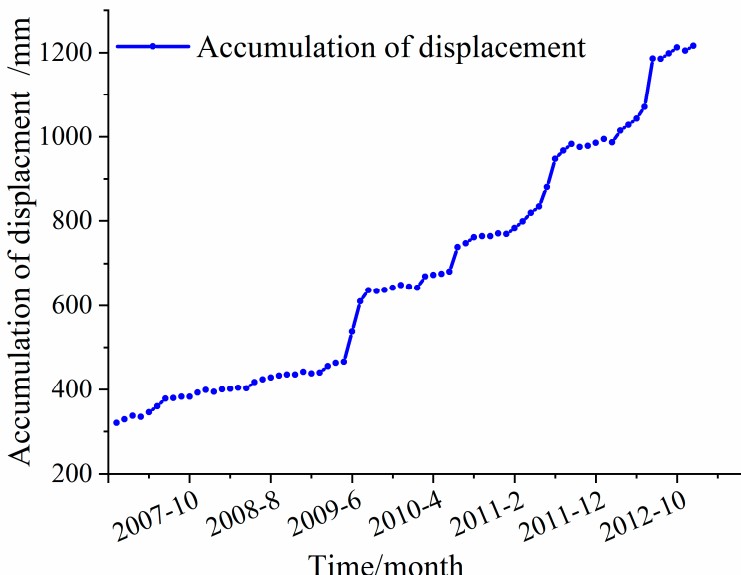

**Figure 9.** The Accumulative displacement curve of the step type landslide.

### 3.2. Landslide Displacement Dataset

To evaluate the effectiveness of the landslide displacement prediction method proposed in this paper, the long-term monitoring dataset of the "Step-type" landslide deformation monitoring stations in the Three Gorges Reservoir area provided by the National Cryosphere Desert Data Center (NCDC) was used to conduct experimental comparative analysis [44,45]. The dataset of the monitoring stations contains cumulative landslide displacement values of the monitored area, rainfall, and reservoir water level values of the Three Gorges Reservoir area. The monitoring GPS data is obtained from the monitoring stations DX01, DX02, DX03, DX04, ZG93, ZG111, and ZG120 in the Baishuihe and Bazimen areas of Zigui County in the Three Gorges Reservoir. The data collection time is from 1 January 2007 to 31 December 2012, in which the time interval of cumulative displacement data collection is one month, and the time interval of rainfall and reservoir water level collection is one day. It has a high missing rate for the rainfall data in the dataset. Therefore, to build a more accurate prediction model, the missing rainfall data is filled by taking the mean value of adjacent data. Through the gray correlation analysis of the pre-processed data set, it can be found that the rainfall data after the data complementation has a higher correlation degree with the periodic displacement component, which is helpful in improving the training accuracy of the prediction model.

It can be seen that the size of these datasets is very small, and it is difficult to fully train the deep learning model through these basic monitoring data. Other methods or mechanisms must be adopted to solve the problems of big data required for the training of the deep learning model under the condition of small sample sets.

## 4. Model Implementation

### 4.1. Decomposition of Landslide Displacement

In this paper, the DDM mentioned in Section 2.1 is used to decompose the cumulative displacement of the landslide, which is divided into the trend term displacement component and periodic term displacement component.

As shown in Figure 10, taking the monitoring data of the ZG110 monitoring station as an example, this paper compares the DDM with the traditional cumulative displacement decomposition method, the Empirical Mode Decomposition (EMD) method. Figure 10a,b respectively show the extraction results of cumulative displacement via two methods. It can be clearly seen that the trend displacement component obtained by the DDM can better capture the physical characteristics of the displacement trend change, which is more in line with the trend changes of cumulative displacement. At the same time, the periodicity of the periodic displacement component obtained by the difference decomposition method is more significant, which is conducive to constructing the prediction model of the periodic component of the landslide in the later stage.

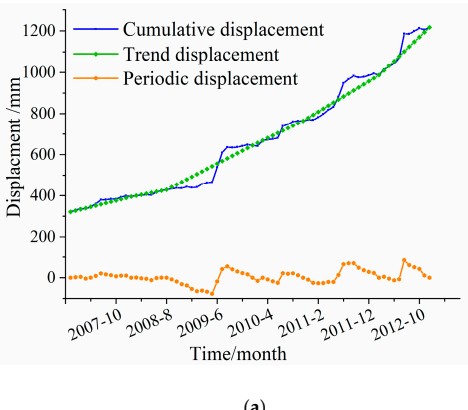
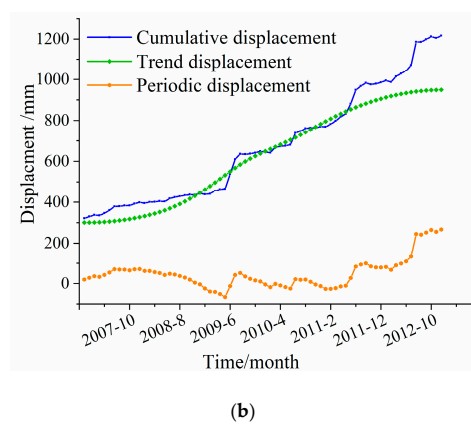

(**a**)          (**b**)

**Figure 10.** Comparison of decomposition results of landslide cumulative displacement by different methods. (**a**) DDM Difference decomposition method. (**b**) EMD decomposition method.

This paper decomposes the cumulative displacement of DX01, DX02, DX03, DX04, ZG93, ZG111, and ZG120 by DDM to obtain several samples ($\leq 72$) of each monitoring station. The trend components and periodic components of the above GPS stations obtained by DDM are shown in Figure 11. Figure 11a shows the trend displacements of all stations, and Figure 11b shows the period displacements. We can see that with the passage of time, the trend displacement of all nodes increases gradually and generally has an obvious upward trend from June to August each year, while the periodic term displacement of all stations shows a strong periodic curve, and the fluctuation frequencies of all nodes are almost the same. Therefore, in the subsequent data enhancement step, there will be more valid data to choose from.

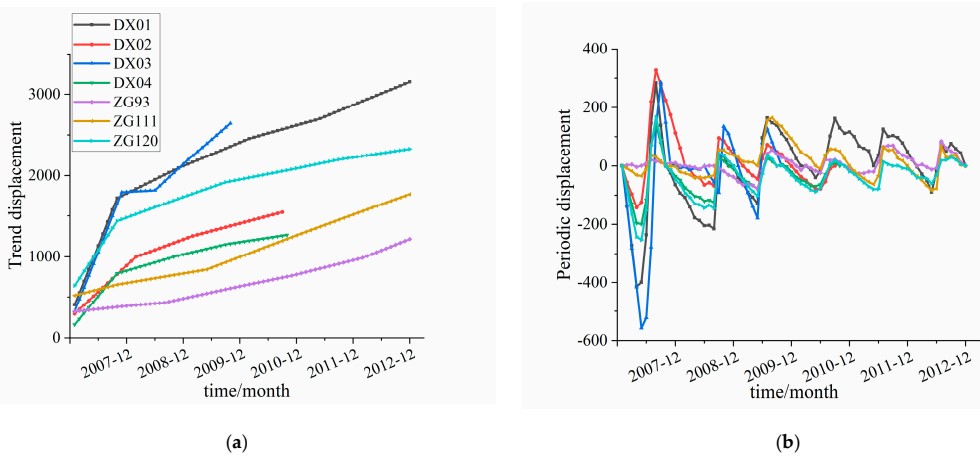

(**a**)                                    (**b**)

**Figure 11.** Trend and periodic displacement of all GPS stations decomposed by DDM. (**a**) Trend displacement. (**b**) Periodic displacement.

### 4.2. Selection of the Inducing Factors of the Periodic Displacement

Since the external inducing factors or influence factors are very important to the prediction accuracy of landslide displacement periodic components, it is necessary to screen out the influencing factors that strongly correlate with the periodic displacement component as the input of the displacement prediction model. Studies show that the influencing factors leading to periodic deformation mainly include rainfall, periodic reservoir water level fluctuations, and landslide deformation state [46].

Among them, rainfall is an important factor causing landslide deformation. On the one hand, rainfall changes the structure of the slope body by scouring the slope surface; on the other hand, the bulk density of the slope body, the strength parameters of sliding zone soil, and hydrostatic pressure are changed through infiltration [47]. Since the change of slope structure by rainfall is a relatively slow process, the impact of current effective rainfall on landslide displacement may lag. Therefore, in this paper, four candidate factors are selected: the cumulative rainfall of the current month *f1*; the maximum rainfall of the current month *f2*; the cumulative rainfall of the previous two months *f3* and the cumulative rainfall of the previous quarter *f4*.

For the influence factor of periodic reservoir water level fluctuations, on the one hand, the rise and fall of the reservoir water level will affect the physical and mechanical properties of the rock and soil through the loading and unloading effects of the dry and wet cycle of the slope. On the other hand, it will affect the slope's internal and external mechanical outcomes by changing the seepage field inside the slope [48]. Therefore, the average water level of the reservoir in the current month *f5*, the monthly variation of reservoir water level *f6*, the variation of the previous two months *f7* and the variation of the reservoir in the previous quarter *f8* are selected as candidate influencing factors.

Under the same external conditions, the displacement responses of landslides in different deformation states are different. When the landslide is in a stable state, the drastic change in the external environment will not cause large-scale changes in the landslide

displacement. When the landslide is in a critical state, a slight disturbance of external factors may destroy the original balance of the landslide system, resulting in abrupt changes in landslide displacement [5]. Therefore, the cumulative displacement of the previous month *f9*, the displacement increment of the previous month *f10*, the displacement increment of the first two months *f11*, and the displacement increment of the previous quarter *f12* are selected as the candidate factors that affect the periodic displacement component.

In this paper, the correlation degree between the above candidate influencing factors *f1~f12* and the displacement of the landslide period term is calculated by the grey correlation analysis method, and the grey correlation values of each feature are shown in Figure 12. The coefficient threshold is taken as 0.7; that is, when the gray correlation degree exceeds 0.7, it is determined that the correlation between the influence factors and the displacement of the periodic term is relatively high. The environmental impact factors with a high correlation with periodic displacement can be obtained through screening: current month rainfall *f1*, previous two-months cumulative rainfall *f3*, current month reservoir water level variation *f6*, two-months reservoir water level variation *f7*, last month's displacement change *f10* and the displacement increment of the first two months *f11*. The specific correlation values of the candidate impact factors are shown in Table 1. It can be seen that the greatest impact on periodic displacement is the cumulative displacement change in the first two months, and the correlation coefficient is 0.81. And whether it is rainfall, reservoir level changes, or cumulative displacement changes, the correlation between the cumulative values of the two months is a little higher.

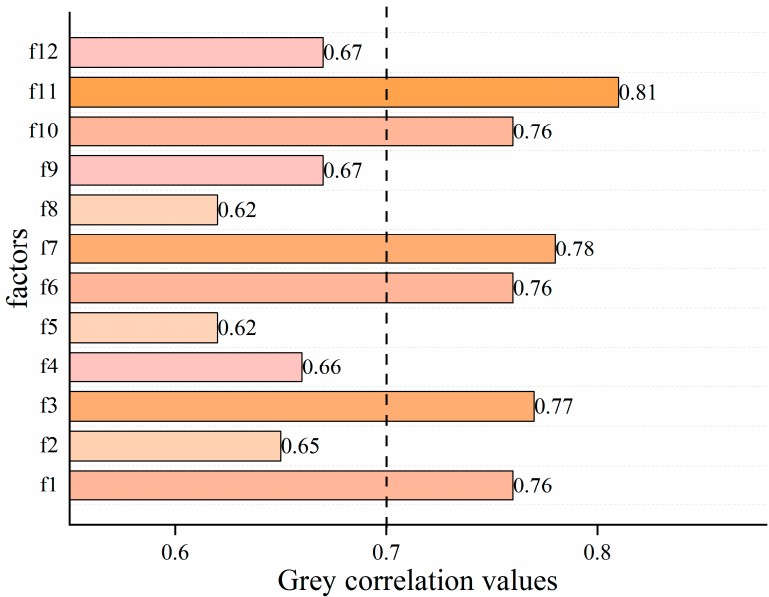

**Figure 12.** Grey correlation values of each environmental factor.

**Table 1.** Grey correlation values between candidate influence factors and displacement components of the periodic term.

| Influencing Factor | Rainfall of the Current Month (f1) | Rainfall of the Previous Two Months (f3) | Reservoir of the Current Month (f6) | Reservoir of the Previous Two Months (f7) | Displacement Change of Current Month (f10) | Displacement Change of Previous Two Months (f11) |
|---|---|---|---|---|---|---|
| Correlation value | 0.76 | 0.77 | 0.76 | 0.78 | 0.76 | 0.81 |

Figure 13 shows the relationship between influencing factors of rainfall (a), reservoir water level (b), landslide deformation characteristic (c), and landslide displacement of the periodic term. Among them, the rainfall factors include the cumulative rainfall values of

the current month and the previous two months, the influencing factors of reservoir water level change include water level change values of the current month and the previous two months, and the deformation characteristics of landslide include cumulative displacement increment values of the current month and the previous two months. It can be seen from Figure 13a that rainfall factors, especially the two-month cumulative rainfall, are highly similar to the fluctuation rule of periodic displacement. From Figure 13b, it can be drawn that the changes in the two-month cumulative reservoir water level lead to periodic displacement changes. In addition, from Figure 13c that monthly and two-month cumulative displacement increments are basically consistent with the variation trend of the periodic displacement component.

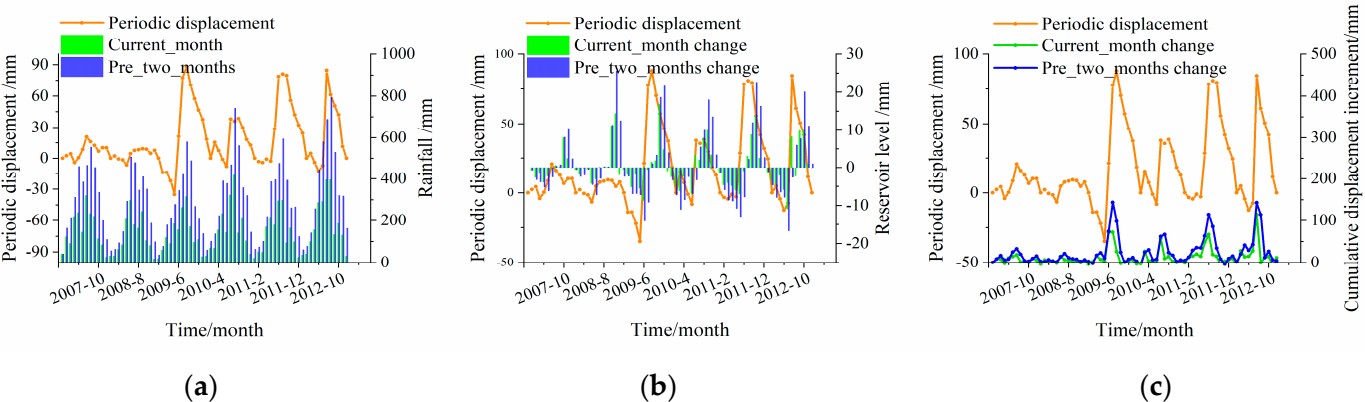

**Figure 13.** Relationship between influence factors and periodic term displacement. (**a**) Rainfall factor. (**b**) Reservoir water level change. (**c**) Cumulative displacement increment.

The fluctuation rules of the above influencing factors are similar to that of the periodic displacement component, and the grey correlation degree value exceeds 0.7. Therefore, in this study, these six factors are selected for the prediction modeling of landslide periodic term displacement. Complete and abundant multivariate influence factors can effectively improve the prediction accuracy of the periodic displacement component prediction model.

### 4.3. Feature Engineering of Periodic Displacement Based on Supervised Learning

In this paper, the dataset of the periodic prediction model is constructed due to the supervised learning method. For the whole data of all GPS stations, the influence factors selected by 4.2 are taken as the feature data, and the corresponding periodic displacement components are used as tags to get the feature-label data. Furthermore, to pay more attention to the time series correlation of periodic displacement, the time sliding window is 12. That is, the influence characteristics and labels of the first 12 steps affect the periodic displacement at the current time.

Figure 14 is the correlation relationship calculated by the Grey Correlation analysis method between the historical periodic displacement and the current periodic displacement of target dataset ZG110, and it can be seen from the figure that the longer the historical interval, the smaller the periodic displacement correlation. Therefore, choosing a historical step of 12 in this paper can learn the periodic displacement timing dependencies in the past year well while excluding unnecessary interfering information.

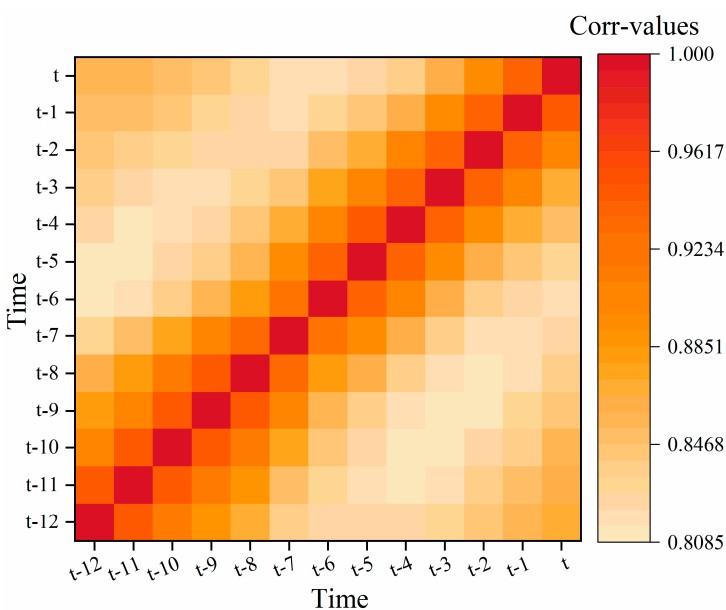

**Figure 14.** Periodic displacement grey correlation heatmap.

*4.4. Data Preparation*

Based on the data set selection, component partition, external influence factor selection, and periodic component feature engineering mentioned above, data enhancement is carried out to enhance the training accuracy and universality of the small sample model. the process of data enhancement is as follows:

1. Construction of original base dataset and target dataset

To fully train the prediction model of "Step-type" landslides, this paper needs to construct a base dataset with a relatively large amount of samples by data supplement, and the enriched base pre-training dataset needs to have similar physical properties to the target predicting dataset to mine much about the spatial-temporal correlations of the similar "Step-type" landslides.

The Baishuihe landslide, Bazimen landslide, and Xintan landslide of the Three Gorges Reservoir area adopted in this paper belong to the same "Step-type" landslide deformation. This paper chooses ZG110 as the target predicting dataset and the other stations like DX01, DX02, DX03, DX04, ZG93, ZG111, and ZG120 are selected as similar candidate datasets to do the further base dataset enhancement.

2. Enhance Robustness of original base and target datasets

To enhance the stability and robustness of the base dataset and target dataset obtained by the fusion processing, a data enhancement algorithm based on the DTW algorithm is used to do further enhancement. The parameters of the DTW algorithm are selected as follows:

The number of selecting epochs $T = 4 * shape$ (for example, when the shape of the base dataset is 359, then the epochs $T_{Base} = 1436$, and if the sample size of the target dataset is 66, then $T_{Target} = 264$). The size of query sequence $m = 30$, the nearest neighbor sequence number $n = 5$. Weights allocated to the Sequence are $w_c = 0.5$, $w_1 = 0.15$, $w_2 = 0.15$, $w_3 + w_4 + w_5 = 0.2$. Where, $w_c$ is the weight value assigned to the central sequence, $w_1$ to $w_5$ are the weight values of screened-out sequences of the five nearest neighbors.

The samples of the base dataset after enhancement based on the DTW algorithm are expanded to 1795 and the samples of the target dataset are expanded to 330. Due to the dataset expansion based on the observation station data with similar landslide type characteristics in the same area and the use of the DTW algorithm for dataset enhancement, the dataset after the data expansion and enhancement not only dramatically expands in

sample size but also has higher stability, which can make prediction model fully trained during the training process and obtain better prediction results.

To eliminate the influence of data dimension, the base and target datasets of the periodic term are normalized to [0, 1]. Then, the 1795 base datasets are divided into three parts: training set, validation set, and test set. Among them, the ratio of training set data to validation set data is approximately 5:1; that is, the amount of the training dataset is 1257, the amount of the validation dataset is 526, and the amount of the testing dataset is 12. At the same time, the 330 target data set are also divided into the same three-part, the sample number of the training set is 318, the sample size of the validation set is 20% of the training set, and the sample number of the testing set is 12.

## 5. Discussion

### 5.1. Prediction of Trend Displacement Component on ZG110

Given the landslide trend component data obtained by DDM, the trend displacement samples from January 2007 to December 2011 of the target dataset ZG110 are used as the model training dataset, and the monitoring data from January 2012 to December 2012 of ZG110 are used as the prediction data. The cubic function fitting method is used to model the training dataset, and the fitted cubic function model is used to predict the future trend data. The cubic function fitting method is shown in Formula (18), where $a_1$, $a_2$, $a_3$, and $a_4$ are the coefficients of the fitting function, and $y$ is the predicted value of trend displacement. Figure 15 shows the fitting and prediction process of the displacement trend component of ZG110 monitoring point data.

$$y = a_1 x^3 + a_2 x^2 + a_3 x + a_4 \tag{18}$$

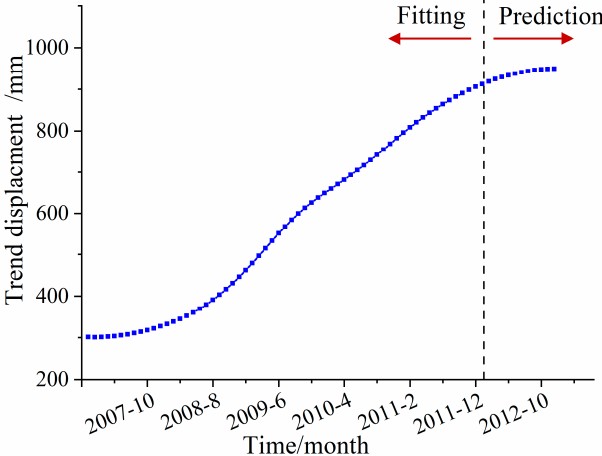

**Figure 15.** Fitting and prediction process for trend displacement on ZG110.

After the cubic fitting calculation of the displacement trend component of the ZG110 monitoring point data, the cubic fitting parameters' values of $a_1$, $a_2$, $a_3$, and $a_4$ are as follows Formula (19), and the correlation coefficient ($R^2$) of the fitting result is 0.98.

$$y = 0.00075 x^3 + 0.021 x^2 + 7.234 x + 313.31 \tag{19}$$

Then, the cubic fitting function obtained by Formula (19) is applied to the prediction of the landslide displacement trend component during the period from January 2012 to December 2012 at the ZG110 monitoring station, and the prediction results are shown in Figure 16. After calculation, it can be obtained that the average absolute error of the trend displacement component prediction result is 8.358 mm, the standard deviation is 4.48 mm, the minimum error value is only 1.13 mm, and the maximum error value is 11.86 mm. The prediction experiment results show that the trend displacement component obtained by the

difference decomposition method has a high degree of matching with the cubic function curve, and the changing trend is more predictable, so more accurate prediction results can be obtained.

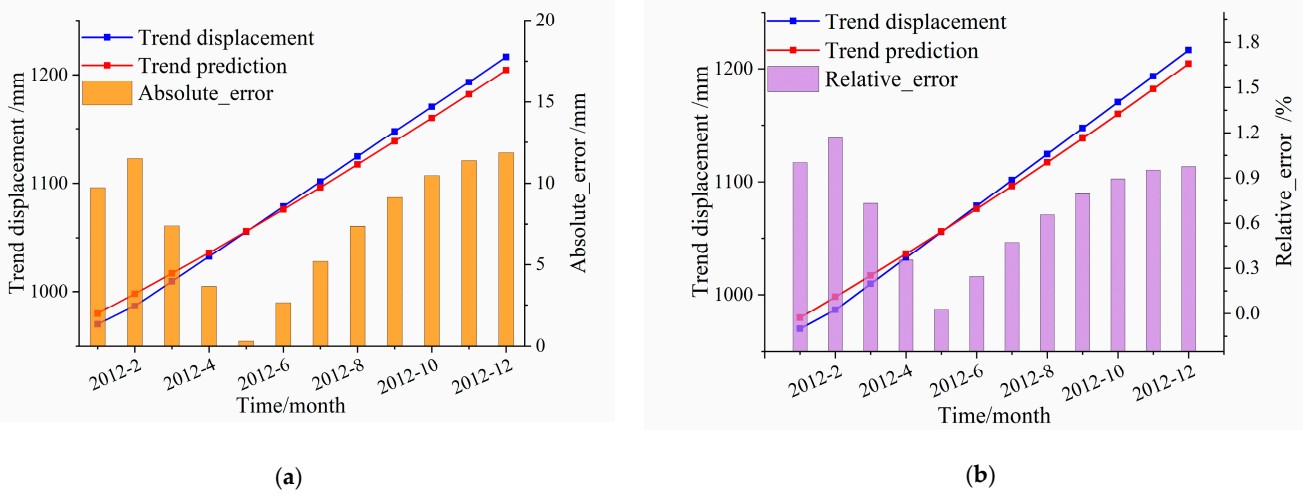

(**a**)　　　　　　　　　　　　　　　　　　　　　　　　(**b**)

**Figure 16.** Prediction results of the trend displacement on ZG110. (**a**) Absolute error. (**b**) Relative error.

### 5.2. Prediction of Periodic Displacement Component on ZG110

The periodic component displacement can be obtained by removing the trend component displacement from the cumulative landslide displacement. The calculation equation of periodic component displacement component $\eta(t)$ is shown in Formula (20).

$$\eta(t) = S(t) - \varnothing(t) \tag{20}$$

where $S(t)$ represents the cumulative landslide displacement and $\varnothing(t)$ represents the trend displacement component. The periodic displacement component of the ZG110 monitoring station obtained by the DDM is shown in Figure 17. It can be seen that the obtained periodic displacement component has relatively stable periodic characteristics. In this paper, the periodic displacement data from January 2007 to December 2011 are used as training data, and the data from January 2012 to December 2012 are used as testing data for comparative analysis.

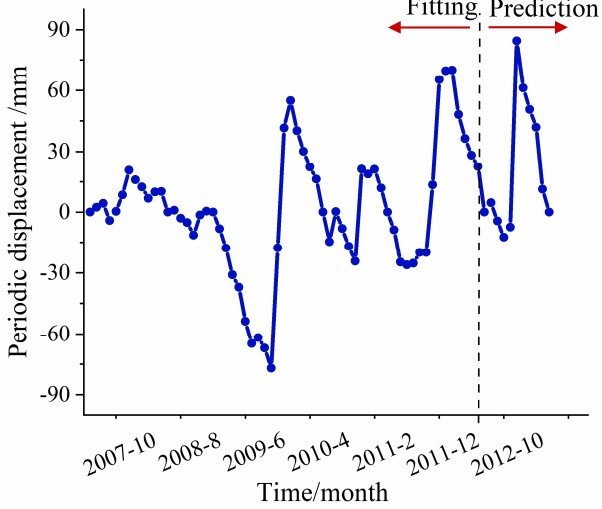

**Figure 17.** The periodic displacement component of the ZG110 monitoring station was obtained by the differential decomposition method.

1. Experimental environment and parameter selection of the TC-DLDPM model

The whole experimental environment of this paper is as follows: the CPU is AMD Ryzen7 5800H, the number of cores is 8, the logical processor is 16, and the GPU is the NVIDIA GeForce RTX 3050 Laptop (4096 MB). The programming language is Python, while the deep learning framework is Keras with Tensorflow2.0 (Google, America) as the back end.

Each sample in the dataset is divided into 12 sub-sequences according to historical time steps. For each sub-sequence, set the convolutional dimension size of the CNN network to 12; the convolutional kernel size to 2; the maximum pooled kernel size to 2; the number of hidden layer neurons in the LSTM network to 200; the loss function of the model to MAE; the learning rate to 0.001; the model training rounds are initially set to 500, and the early stop function is set when the loss obtained by the model on the validation set data does not drop in 20 times. The whole training and prediction process of the model was repeated 50 times. Finally, the average prediction results and the average error index of the model were compared and analyzed with other methods.

2. Comparison and analysis of the prediction results of the periodic term displacement components

Firstly, the base dataset is thoroughly learned and trained based on TC_DLDPM in stage 1 (basic training), and the corresponding model parameters of each layer are obtained to establish the base displacement prediction model. Then, the parameters of each network layer obtained in the first stage are migrated to the small sample fine-tuning modeling process based on the target dataset in the second stage. The parameters of layers 4 to 9 of the model, that is, Time-step&features Attention Block layers' parameters, are updated on the training set of the target dataset with 1/20 of the original learning rate (i.e., $5 \times 10^{-5}$). Finally, the displacement component data of periodic term of ZG110 monitoring station test set from January 2012 to December 2012 are predicted, and the prediction results of periodic displacement component are obtained.

The displacement prediction result of the period term of the ZG110 test set and its absolute error are shown in Figure 18. (1) From January to March 2012, the absolute errors of the predicted displacement component of the periodic term are relatively small, and the predicted results are highly consistent with the real displacement values. Among them, the prediction error of February is relatively large, which is 5.76 mm. (2) From March to May, the displacement of the periodic term showed a slow downward trend, and the displacement error gradually increased. It rose from 2.30 mm in March to 8.89 mm in May. (3) In the landslide disaster-prone period from May to July, the reservoir's water level decreases to 146.90 m, and the landslide displacement of the periodic term rises in Step-type due to the sudden increase in the monthly rainfall and accumulated rainfall. The minimum error occurred in June, and the error value was 6.67 mm. The maximum error occurred in July, and the error value was 14.50 mm. It shows that the prediction model still has a strong prediction ability when the external influence factors change rapidly, and the comprehensive prediction effect is ideal. (4) From August to December, the reservoir's water level rose to 174.13 m, the rainfall gradually decreased, and the displacement value of the period term steadily decreased. During this period, the prediction error of the model is tiny, very close to the actual value, the maximum error is 8.56 mm, and the minimum error is 0.16 mm. During the entire test period, the mean absolute error (MAE) and correlation coefficient $R^2$ of the prediction model are 5.66 mm and 0.95, respectively, indicating that the dataset with data expansion and enhancement not only greatly expands the sample size but also has higher stability, which can enable the prediction model to be fully trained during the training process, and the overall prediction accuracy has been improved.

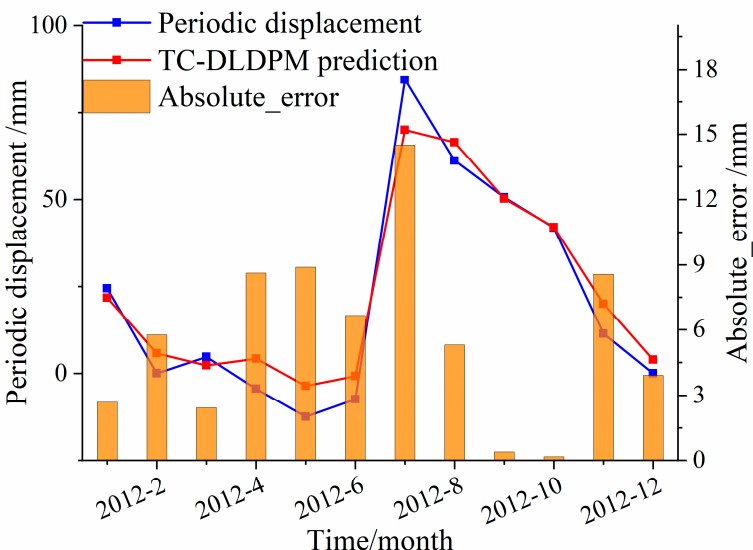

**Figure 18.** Displacement prediction results of periodic term on the ZG110 testing data.

To verify the performance of the TC-DLDPM model, the Random Forest (RF) model, Support Vector Regression (SVR) model, error Back Propagation (BP) network, Long and Short-Term Memory (LSTM) network, Gate Recurrent Unit (GRU), Bi-directional long-short term memory (Bi-LSTM) [49] and the CEEMDAN model [50], which are applied to the field of landslide displacement prediction are selected to compare the prediction accuracy in this study. The target datasets of all algorithm models are the periodic displacement data obtained by the ZG110 monitoring station of the Bazimen landslide displacement in the Three Gorges Reservoir area and the periodic displacement data from January 2012 to December 2012 are used as the test data. The network parameters of the comparison model are described as following Table 2:

**Table 2.** Parameter settings of the different prediction models.

| Models | Parameters | Values | Models | Parameters | Values |
|---|---|---|---|---|---|
| RF | Number of trees | 100 | GRU | hidden layers | 100 |
| | maximum tree depth | 3 | | Time-sliding window size | 12 |
| | minimum number of samples | 20 | | learning rate | 0.001 |
| | minimum number of leaf nodes | 5 | | batch_size | 64 |
| | maximum number of features | 10 | | training epochs | 500 |
| SVR | penalty factor C | 5.3 | Bi-LSTM | hidden layers | 100 |
| | kernel function parameter Gamma | 0.01 | | Time-sliding window size | 12 |
| BP | hidden layers | 160 | | learning rate | 0.01 |
| | learning rate | 0.001 | | batch_size | 64 |
| | training epochs | 100 | | training epochs | 500 |
| LSTM | hidden layers | 200 | CEEMDAN | hidden layers | 128 |
| | Time-sliding window size | 12 | | Time-sliding window size | 12 |
| | learning rate | 0.001 | | learning rate | 0.001 |
| | batch_size | 64 | | batch_size | 64 |
| | training epochs | 500 | | training epochs | 500 |

Note: The training loss function of the above model is set to 'MAE,' and the training process adopts the same early stop function as TC-DLDPM. All models use GridSearchCV methods to find the best parameters for prediction.

The prediction values of the periodic term displacements of the ZG110 dataset from January 2012 to December 2012 are shown in Table 3 and the monthly absolute error values' heatmap of the test data set is shown in Figure 19. It can be seen in Figure 19 that the experimental results of shallow learning network RF, SVR, and BP networks for periodic displacement prediction are not ideal, especially when the environmental factors

like rainfall (June, July, August) change violently, the prediction error is relatively large; the prediction error of LSTM and GRU networks is smaller than that of shallow neural networks, but the model does not better fit the periodic displacement evolution trend when the environmental factors change greatly; the combined deep learning models Bi-LSTM and CEEMDAN which are specifically targeted for the "Step-type" landslide prediction can fit the evolution of periodic displacement well and have a better prediction. However, the TC-DLDPM proposed in this paper can perform more detailed and accurate training based on pre-training, so the prediction result error is smaller and the coherence coefficient ($R^2$) of the model is greater.

**Table 3.** Comparison of the prediction accuracy and the error of periodic term displacement by the different prediction models.

| Time Step | Real Data | RF | SVR | BP | LSTM | GRU | Bi-LSTM | CEEMDAN | TC-DLDPM |
|---|---|---|---|---|---|---|---|---|---|
| 2012/1 | 24.42 | 23.84 | 20.01 | 12.54 | 25.84 | 24.44 | 21.43 | 19.92 | 21.74 |
| 2012/2 | 0.00 | 8.43 | 19.96 | 17.57 | 6.75 | 4.35 | 0.68 | 4.20 | 5.76 |
| 2012/3 | 4.72 | −0.47 | 14.43 | 9.45 | 12.88 | 7.04 | 6.81 | 0.72 | 2.30 |
| 2012/4 | −4.36 | −0.27 | 14.17 | 12.48 | 5.19 | 1.75 | −9.51 | −1.16 | 4.26 |
| 2012/5 | −12.44 | 2.37 | 15.76 | 4.97 | −1.00 | −1.59 | −17.11 | −9.34 | −3.55 |
| 2012/6 | −7.42 | 6.28 | 24.37 | 0.71 | 5.78 | 4.63 | −5.25 | 13.90 | −0.76 |
| 2012/7 | 84.50 | 31.30 | 62.46 | 32.90 | 82.44 | 76.25 | 70.66 | 72.00 | 70.00 |
| 2012/8 | 61.12 | 58.20 | 62.53 | 64.70 | 68.16 | 64.01 | 86.30 | 54.00 | 66.42 |
| 2012/9 | 50.64 | 58.16 | 54.09 | 61.37 | 62.07 | 58.06 | 72.27 | 54.64 | 50.25 |
| 2012/10 | 41.96 | 54.64 | 44.78 | 42.22 | 51.33 | 50.31 | 58.50 | 44.86 | 42.13 |
| 2012/11 | 11.38 | 44.61 | 30.57 | 53.22 | 36.85 | 30.64 | 17.90 | 8.18 | 19.94 |
| 2012/12 | 0.00 | 9.55 | 25.73 | 26.45 | 27.00 | 23.27 | 0.46 | 10.10 | 3.92 |
| **MAE/MM** | | 13.82 | 15.60 | 17.59 | 11.07 | 8.76 | 8.49 | 6.68 | 5.66 |
| **$R^2$** | | 0.56 | 0.61 | 0.61 | 0.80 | 0.87 | 0.91 | 0.93 | 0.95 |

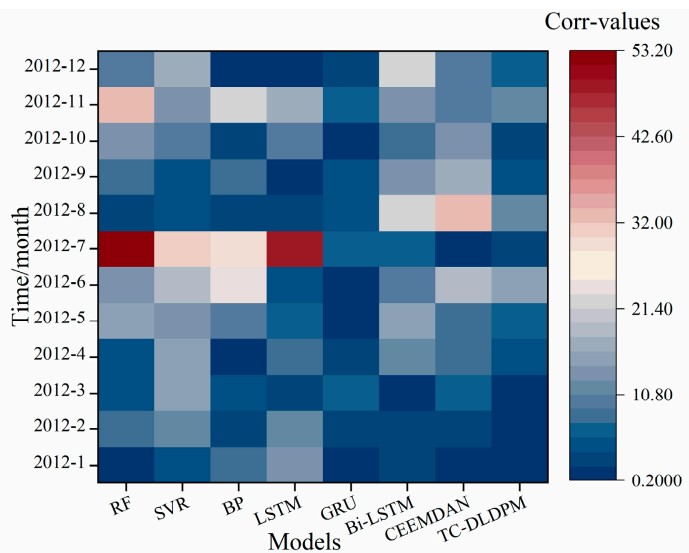

**Figure 19.** Prediction MAE of Different methods in ZG110 testing set.

By comparing the simple deep learning model Bi-LSTM and the combined deep learning model CEEMDAN which have better prediction precision than the TC-DLDPM model proposed in this paper on the ZG110 testing set, the prediction fitting results can be seen in Figure 20. Among the three prediction methods, the fitting result of Bi-LSTM is the worst. Combined with the error analysis in Table 3, the MAE error of the Bi-LSTM is 8.49 mm; the prediction results fit well with the real data from January to June 2012, but from July to October, when the rainfall and reservoir water level change greatly, the prediction error effect is very poor, and the maximum absolute error occurs in August which reaches to 25.18 mm. In addition, there is a lag in the prediction of the fluctuation

trend of periodic landslide displacement, which indicates that a single Bi-LSTM model is still greatly influenced by the external environment when predicting the complex nonlinear evolution process of landslide displacement. In addition, because RNN iterates in time order, the error of Bi-LSTM will be superimposed gradually, resulting in a large fitting deviation. While the fitting result of the CEEMDAN combination model is relatively better and the average error of the prediction for the whole year of 2012 is only 6.68 mm, which is better fitted with the real data, and the fluctuation trend of periodic displacement is also well fitted, but similarly, there is still a large error in the rainy season from June to August 2012, the maximum prediction error occurs in June which is 21.32 mm. The TC-DLDPM model proposed in this paper achieves much better results in both the fitting of the fluctuation trend and the prediction of monthly displacement. The overall MAE error is only 5.66 mm, and it has a better prediction effect no matter how the external factors such as rainfall and reservoir water level change. It is because this method has learned the common characteristics of a large number of landslides of the same type in advance and then focuses more on the characteristics of their own time series and environmental factors, so it has better prediction accuracy.

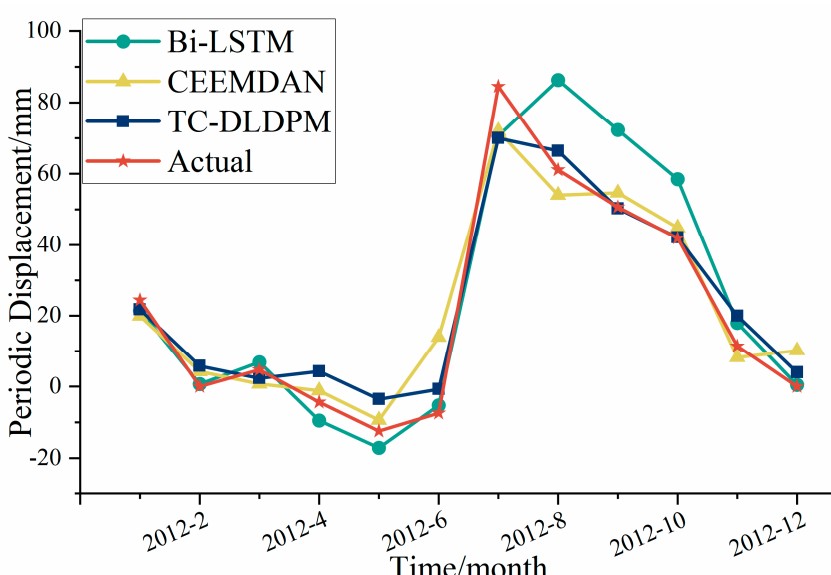

**Figure 20.** Comparison of the prediction results of the periodic term displacement by different models.

### 5.3. Prediction Experiments of the Landslide Cumulative Displacement

According to the time series decomposition principle of DDM mentioned in Section 2.1, the predicted values of trend displacement and periodic displacement are added together to obtain the predicted value of the cumulative landslide displacement. The cumulative displacement prediction experiment of the ZG110 monitoring station during the test period, i.e., January 2012 to December 2012, was carried out. The prediction accuracy and error analysis of cumulative displacement are shown in Table 4 and Figure 21 shows the accuracy comparison results between the predicted value of the cumulative displacement and the actual observed value. It can be seen from the figure that the predicted value of cumulative landslide displacement is highly fitted with the actual value, and the average error of prediction is 8.92 mm. Due to the sudden change in rainfall which is a key external influence factor, the maximum error occurred in July 2012, with an error value of 19.71 mm. In general, the prediction accuracy of cumulative landslide displacement is relatively high, and the step trend of the cumulative landslide displacement can be well predicted.

**Table 4.** Prediction value and error analysis results of the ZG110 cumulative displacement.

| Time Step | Trend Predicted Value (mm) | Periodic Predicted Value (mm) | Cumulative Predicted Value (mm) | Cumulative Real Value (mm) | Absolute Error (mm) | Relative Error (%) |
|---|---|---|---|---|---|---|
| 2012/1 | 979.60 | 21.74 | 1001.34 | 994.30 | 7.04 | 0.71 |
| 2012/2 | 998.10 | 5.76 | 1003.86 | 986.60 | 17.26 | 1.75 |
| 2012/3 | 1016.97 | 2.30 | 1019.27 | 1014.30 | 4.97 | 0.49 |
| 2012/4 | 1036.22 | 4.26 | 1040.47 | 1028.20 | 12.27 | 1.19 |
| 2012/5 | 1055.85 | −3.55 | 1052.30 | 1043.10 | 9.20 | 0.88 |
| 2012/6 | 1075.87 | −0.76 | 1075.11 | 1071.10 | 4.01 | 0.37 |
| 2012/7 | 1096.29 | 70.00 | 1166.28 | 1186.00 | 19.72 | 1.66 |
| 2012/8 | 1117.11 | 66.42 | 1183.53 | 1185.60 | 2.07 | 0.18 |
| 2012/9 | 1138.34 | 50.25 | 1188.59 | 1198.10 | 9.51 | 0.79 |
| 2012/10 | 1159.98 | 42.13 | 1202.11 | 1212.40 | 10.29 | 0.85 |
| 2012/11 | 1182.05 | 19.94 | 1201.99 | 1204.80 | 2.81 | 0.23 |
| 2012/12 | 1204.54 | 3.92 | 1208.46 | 1216.40 | 7.94 | 0.65 |
| *Maximum error* | | | 19.72 mm | | | |
| *Minimum error* | | | 2.07 mm | | | |
| *Average error* | | | 8.93 m | | | |

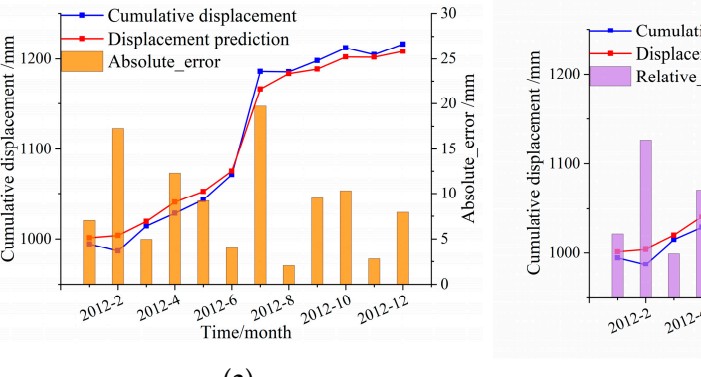
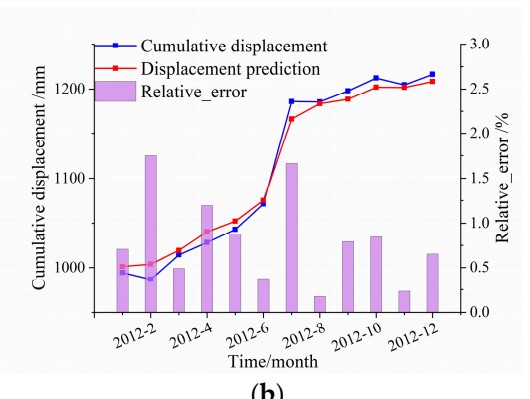

(**a**)　　　　　　　　　　　　　　(**b**)

**Figure 21.** Accuracy comparison results between the predicted value of cumulative displacement and the actual observed value. (**a**) Absolute error. (**b**) Relative error.

## 6. Conclusions

Aiming at the in-depth research and analysis of landslide displacement under the condition of small samples, a two-stage combined depth learning dynamic prediction model TC-DLDPM is proposed in this paper to realize the comprehensive prediction task of landslide displacement.

The dataset was expanded through the DTW algorithm according to the similarity of multiple stations monitoring data, the geographical environment, and the Step-type of landslides. Then, the TC-DLDPM model captures the common law between input and output and realizes transfer learning on small sample data through the two combined stages of Basic model learning and Small sample fine-tunning. The proposed model has been applied to Bazimen and Baishuihe landslides in the Three Gorges Reservoir, and the experimental results demonstrated it could adapt well to the condition of a small sample in the application area. The TC-DLDPM model also has been proved that can shorten the prediction time and improve the prediction accuracy. Thus, the study of the TC-DLDPM model provides a better solution and exploration idea for landslide displacement prediction under the condition of small samples.

In future work, the time series InSAR data and other multi-source monitoring remote sensing data will be introduced to analyze the landslide, and the actual observation of GPS data will be verified and analyzed to construct a more stable, accurate, and efficient landslide deformation warning model.

**Author Contributions:** Conceptualization, C.Y. and J.H.; methodology, C.Y.; software, C.L.; validation, C.Y., J.H. and C.L.; resources, J.H. and Y.Z.; writing—original draft preparation, C.Y.; writing—review and editing, J.H.; visualization, C.L.; supervision, J.H.; project administration, J.H. and Y.Z.; funding acquisition, J.H. All authors have read and agreed to the published version of the manuscript.

**Funding:** This work is supported by the National Nature Science Foundation of China (Grant No. 61862038), "Double-First Class" Major Research Programs, Educational Department of Gansu Province (No. GSSYLXM-04), Gansu Province Science and Technology Program—Innovation Fund for Small and Medium-sized Enterprises (21CX6JA150), Data Project of National Cryosphere Desert Data Center (NCDC) and the Foundation of a Hundred Youth Talents Training Program of Lanzhou Jiaotong University.

**Data Availability Statement:** The dataset is provided by National Cryosphere Desert Data Center. (http://www.ncdc.ac.cn, accessed on 1 September 2021).

**Conflicts of Interest:** The authors declare no conflict of interest.

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
