# Peer review of "Landslide Displacement Prediction Based on a Two-Stage Combined Deep Learning Model under Small Sample Condition"

_remotesensing, doi:10.3390/rs14153732_

Round 1
Reviewer 1 Report
Dear authors,
I read your paper. It is extensive and very detailed. In writing future papers, try to be more concise. I noticed some little things through the paper that I point out to you in detail.
Line 63. Maybe it is a better to use term complex instead complicated.
Line 153 capital letter at the beginning of the sentence.
Figure 4 letters Wf, Wi etc. should be visible in full.
Figure 7 In the picture, it is necessary to indicate the study area more clearly, also in better quality.
Line 837 a dot appears where there should be a space between words.
All tables displayed throughout the paper must be written in the same style.
Table 5 try reducing the number of decimal places in the displayed numbers. Try to keep the same style when writing numbers throughout the text.
Please, harmonize the way of citing references both through the text and through the list of references. If possible, add http address and/or doi number.
Reviewer 2 Report
The topic of this manuscript is interesting, but I struggled with reading this paper to understand the methods and outcomes. I believe that some sentences should be revised (e.g., lines 33-35) and the use of verb tenses as well (e.g., lines 75). Therefore, my advice is a super-major revision to have a scientific presentation of the results for the community and any possible use of the approach by the interested readers.
Section 1 needs to be cited by the most representative references. All detailed information about the methods should be presented in Section 2. Therefore, remove them (e.g., Section 5.1), please. Detailed information is presented in Section 2. You can refer to some highly cited papers for explaining the methods, but scientific explanations of the key expressions should be presented in detail.
The number of the figures is high, for example, it is better to have Figure 7 and Figure 8 in one plot with the same scale bar (i.e., kilometer). Please combine Figure 13, Figure 14 and Figure 15, and add titles including (a), (b) and (c) for the figure with a strong discussion inside the text.
In addition, the figures’ caption needs to be rewritten, for example in Figure 1, the text needs to be replaced with more information about the algorithm. It is also true for the tables’ caption. Each title of the figures should be explained in the caption (i.e., (a), (b)), and discussed the results. There are no titles for the vertical and horizontal axis (i.e., x and y) in some figures (e.g., the title of x in Figure 22). The legend is not informative in some figures, for example in Figure 19, so the legend needs to be removed.
The abstract section definitely needs to be rewritten.
Lines (12-17): Please add some explanations about your physical problem, i.e. landslide, after that try to convince the readers that your proposed approach is able to overcome challenges that other methods have not been able in the previous studies.
Line (15): What is CNN? Surely, it is an abbreviation. You should present full names before applying their abbreviation.
Lines (20-21): I strongly ask you to rewrite this sentence.
Lines (21-23): It is not an excellent way to present your results for scientific publishing.
Lines (23-25): The conclusion is not strong enough with some grammar issues.
Lines (53-55): This sentence is not a great motivation about the significance of your work.
Lines (73-75): Please complete this part by citing the state-of-the-art techniques from data-driven tools.
Line (156): Please add a short explanation of all the methods with a figure, as an overall representation of the procedure.
Line (466): Please add, “in”
Lines (706-707): Please remove the table. The parameters are presented in the equation without a number. Please add numbers to all the equations and cite them inside the text by their number.
Lines (802-809): I highly recommend rewriting these explanations, this manuscript is a scientific paper!
In the conclusion section, future directions should be highlighted.
Reviewer 3 Report
The paper presents the Two-stage Combined Deep Learning Dynamic Prediction Model (TC-DLDPM) under small sample condition to solve the problem of insufficient training of displacement prediction model due to the small number of monitoring samples. So the scope of the paper is really actual. The proposed approach is novel. The topic suties to need of Remote Sensing journal.
My recommendations to this paper:
$1 Introduction is very long. Thus I recommend introducing the subsections. It will simplify the way of following it.
$2 At the end of introduction please add a paragraph with organization of the paper
$3 Between 2. Materials and Methods and 2.1. Difference decomposition method(DDM) for landslide displacement decomposition please add the general introduction to section 2.
$4 Qualit of figure 1 must be improved. Now the text "v(t)<v(t-1) and v(t)<v(t+1)?" is out of block.
$5 For section 3 Study area and the landslide displacement data set and section 4 4. Model implementation I have the same recommendation like $3.
$6 Please add the information about what correlation coefficient was selected - Pearson? Spearman?
Round 2
Reviewer 2 Report
The requested corrections have been made and are acceptable
Reviewer 3 Report
The authors responded in correct way my previous review.
Thus I recommend to publish this article in present form.
Best regards,
dr Michał Jasiński
Wroclaw University of Science and Technology.